# Promising and Minimally Invasive Biomarkers: Targeting Melanoma

**DOI:** 10.3390/cells13010019

**Published:** 2023-12-20

**Authors:** Pavlina Spiliopoulou, Carlos Diego Holanda Lopes, Anna Spreafico

**Affiliations:** 1Princess Margaret Cancer Centre, University Health Network, Toronto, ON M5G 2C1, Canada; pavlina.spiliopoulou@glasgow.ac.uk; 2School of Cancer Sciences, University of Glasgow, Glasgow G61 1BD, UK; 3Oncology Centre, Hospital Sírio-Libanês, São Paulo 05001-100, Brazil; cdhlopes@gmail.com

**Keywords:** melanoma, minimally invasive biomarkers, circulating tumour DNA, circulating melanoma cells, extracellular vesicles, intestinal microbiome, prognostic, predictive

## Abstract

The therapeutic landscape of malignant melanoma has been radically reformed in recent years, with novel treatments emerging in both the field of cancer immunotherapy and signalling pathway inhibition. Large-scale tumour genomic characterization has accurately classified malignant melanoma into four different genomic subtypes so far. Despite this, only somatic mutations in *BRAF* oncogene, as assessed in tumour biopsies, has so far become a validated predictive biomarker of treatment with small molecule inhibitors. The biology of tumour evolution and heterogeneity has uncovered the current limitations associated with decoding genomic drivers based only on a single-site tumour biopsy. There is an urgent need to develop minimally invasive biomarkers that accurately reflect the real-time evolution of melanoma and that allow for streamlined collection, analysis, and interpretation. These will enable us to face challenges with tumour tissue attainment and process and will fulfil the vision of utilizing “*liquid biopsy*” to guide clinical decisions, in a manner akin to how it is used in the management of haematological malignancies. In this review, we will summarize the most recent published evidence on the role of minimally invasive biomarkers in melanoma, commenting on their future potential to lead to practice-changing discoveries.

## 1. Introduction

The advent of novel cancer immunotherapies, as well as the discovery of driver signalling pathways in melanoma, such as the mitogen activate protein kinase (MAPK) pathway, have brought an unparalleled improvement in the survival of patients with advanced disease [1,2]. Immunotherapy, in the form of immune checkpoint inhibition, is associated with durable response rates and long-term survival but is also characterised by slow kinetics, when compared to targeted treatments. On the other hand, abrogating the MAPK pathway, with small molecule inhibitors, can elicit rapid response and swift symptomatic relief for the patients, which are however not as durable as those observed with immunotherapy and are invariably followed by emergence of resistance.

Melanoma is clinically characterised as a rapidly growing tumour that needs urgent decision making and close patient observation. Measures of response to existing melanoma treatments include, as in other tumour types, the traditional clinical response, as well as radiological response, both of which can contain inherent subjective biases. Specifically radiological responses, as measured by the RECIST 1.1 framework, have been found to poorly correlate with patient survival [3,4], and moreover, novel patterns of disease response emerging with current immunotherapy strategies might require more sensitive readouts. The phenomenon of *pseudo progression* observed with immunotherapy, for example, has no clear predictors and is an indication that molecularly intricate biomarkers will be required to accurately assess responses and to ensure continuation of treatment in cases when robust peritumoural inflammation is wrongly regarded as true disease progression [5]. Novel biomarkers such as measurement of circulating tumour DNA hold promise as a tool to aid distinction between *pseudo progression* and real disease progression [6,7,8,9].

Substantial efforts are directed nowadays towards minimally invasive biomarkers in oncology and specifically in melanoma. These can have considerable benefits for cancer patients, given the inherent reduced human risk involved in their collection and their rapid turnover. This could be particularly pertinent to melanoma, where access to tissue can be restricted due to the small actual size of the primary tissue. Additional aims throughout biomarker discovery should focus on the biomarker’s ability to represent a longitudinal surrogate of tumour burden, tumour clonal divergence and heterogeneity [10,11]. Herein, we provide an overview of circulating and other minimally invasive biomarkers for therapeutic monitoring of patients with melanoma. In particular, the use of nucleic acids, circulating tumour cells, exosomes and intestinal microbiota as surrogate markers for treatment response are explored (Figure 1). Not least, challenges in the application of these markers and opportunities for future work are described.

## 2. Circulating Surrogate Biomarkers in Melanoma before the Next Generation Sequencing (NGS) Era

Circulating molecules can be detected in patients with melanoma, either due to active cellular secretion or as a bioproduct of cell death. Multiple serological protein markers have been explored, with lactate dehydrogenase (LDH) currently being the only surrogate widely used in clinical practice. LDH has a significant role in disease stratification via its incorporation into the tumour, nodes and metastasis (TNM) staging [12]. There is strong evidence that LDH, a ubiquitous enzyme that is critical to anaerobic metabolism [13], correlates with prognosis and treatment response in melanoma patients treated with either targeted treatment or immunotherapy [14,15,16,17,18,19]. However, LDH is not consistently elevated in patients with high burden of disease, and its increase in the serum can simply be a consequence of any cellular necrosis. Therefore, LDH cannot be considered a specific marker for longitudinal treatment monitoring and is not always helpful in guiding treatment decisions in this context [10,20,21,22].

S100 is another important melanoma-specific marker, typically detected via immunohistochemistry to diagnose the presence of metastatic melanoma disease in the clinical setting [23,24]. The S100 protein family consists of 21 functionally different proteins that are aberrantly over-secreted [25], and their detectability in viable melanoma cells, as well as in blood circulation, make them promising markers for melanoma disease monitoring and prognosis [26]. Similarly to LDH, the detection of S100 has been found to correlate with disease activity and prognosis in both small- [24,27] and large-scale patient cohorts [28,29,30]. Therefore, although to a lesser degree than LDH [31], its use in the clinic has been adopted in certain parts of the world [32,33].

To date, no other widely validated, minimally invasive biomarkers are available for melanoma. The limited sensitivity and specificity of LDH and S100 highlight the need to discover molecular surrogates for melanoma biology and evolution. The advent of next-generation sequencing (NGS) and other novel technologies has bolstered the genomic characterisation of melanoma at a granular level and is currently continuously supporting the advancement of a new class of highly specific biomarkers (Table 1).

## 3. Mutational Landscape in Cutaneous and Non-Cutaneous Melanoma

Comprehensive genomic classification of cutaneous melanoma was achieved in 2015 via a systematic multi-platform characterization performed by the Cancer Genome Atlas Network [34], whereby the subdivision of cutaneous melanoma in the four types of mutant *BRAF*, mutant *RAS (N/H/R)*, mutant *NF1* and triple wild-type (triple WT) emerged. Hotspot mutations predominantly in the V600 codon of *BRAF* and the Q61 codon of *NRAS*, as well as loss-of-function events in *NF1*, led to a cascade of perturbations causing upregulation of the MAPK pathway, the most deregulated signalling pathway in cutaneous melanoma [35,36]. Triple WT tumours, which show a higher rate of somatic copy number alterations (CANs), can still exhibit constitutional activation of the MAPK pathway via alternative genomic aberrations like, for example, *KIT* mutations/amplifications [34]. A degree of cell cycle pathway disruption via mutations or deletions in cyclin-dependent kinase inhibitor 2A (*CDKN2A)* has been observed to overlap all four subtypes, whereas mutations of the telomerase reverse transcriptase (*TERT*) promoter are commonly found in the *BRAF/RAS/NF1* mutated subtypes.

Uveal melanoma, which originates from melanocytes residing in the uveal track, has a molecular profile distinct from that of the cutaneous melanoma [37]. Here, we observe large genomic losses, including chromosomal losses in 1p, 6q, 8p and 16q; gains in 6p and 8q; and more importantly chromosome 3 monosomy [38,39,40]. Chromosome 3 aberrations are critically prognostic and, furthermore, contribute to *BAP1* aberrancy via biallelic gene loss when superimposed *BAP1* mutations are also present [37]. Additional mutations in *EIF1AX* and *SF3B1* have been associated with a favourable prognosis subtype of uveal melanoma, whereas mutations in *GNAQ/GNA11* are found in more than 90% of patients [37,41,42]. Lastly, mucosal melanoma also presents a unique molecular profile compared to cutaneous and uveal subtypes. The rate of *NF1* mutations is comparable to that of cutaneous melanoma (~14%); however, *BRAF/NRAS* mutations rates are remarkably lower, and *KIT* mutations are observed in at least 13% of cases [35,43,44].

The growing knowledge on the mutational landscape of melanoma has contributed enormously to the development of some of the minimally invasive biomarkers explored in this review.

## 4. Circulating Tumour DNA in Melanoma

***ctDNA introduction.*** Circulating tumour DNA (ctDNA) is the component of fragmented cell-free DNA (cfDNA) derived from cancer cells. Over the last decade or so, there has been a myriad of studies confirming that detection of ctDNA is readily possible and reproducible in patients with advanced malignancies [45,46,47,48,49,50]. ctDNA-based genomic profiles are highly concordant with those of tumour tissue, and ctDNA levels can directly correlate with tumour burden. Quantitative and digital polymerase chain reactions (like digital droplet PCR (ddPCR)) were the first ctDNA detection approaches for selective gene targets, but more recently, next-generation sequencing (NGS)-based techniques have increasingly been developed [47,48,49,51]. Both methods can be attractive in different contexts. ddPCR is suggested to have higher sensitivity of 0.001% when compared to NGS [52], and furthermore, it has an easy upstream preparation process and faster turnaround time and does not require highly skilled bioinformatics to facilitate analysis. The main drawback of ddPCR is that it can only test for known genomic aberrations, and multiplex biomarker detection is limited. In contrast, NGS allows for the detection of multiple mutations per patient and high sensitivity ctDNA tracking. NGS methods are applied in a “tumour-informed” manner and provide a signature of putative cancer-derived aberrations [53], and it may identify any type of genetic change in all the target regions screened, including novel mutations as well as chromosomal rearrangement and fusion genes. Although this is against the backdrop of being more costly and requiring more complex analytical methods compared to ddPCR [54], NGS methods certainly offer a more comprehensive and personalized approach to ctDNA detection. NGS can be performed via whole-genome sequencing (WGS), whole-exome sequencing (WES), targeted (TS) or candidate gene sequencing (CGS). The lower sensitivity encountered with NGS mandates a higher concentration of ctDNA, which can be challenging in patients with low disease burden [55,56].

Detecting circulating DNA offers multiple opportunities in (a) early cancer diagnosis (b) assessment of minimal residual disease (MRD) (c) disease tracking during anticancer treatment—“molecular response” to therapy and (d) uncovering tumour heterogeneity and novel targets conferring treatment resistance [57,58,59,60,61,62]. Variant allele frequency (VAF) is the most common readout for a mutation detected in ctDNA and is defined as the fraction of cfDNA molecules sequenced at a particular locus that harbours the variant of interest. Several ctDNA assays have now entered clinical practice for various solid tumour types, with the bulk of the evidence stemming from lung cancer studies. In various contexts, such as lung, breast, colorectal and prostate cancer, plasma ctDNA detection assays can complement tissue genotyping and, in combination with tissue sequencing, increase the rate of driver mutation detection [63,64,65,66,67].

***ctDNA in prognostication of advanced melanoma.*** A recent meta-analysis of 19 studies involving patients with metastatic melanoma summarized the role of ctDNA detection at baseline, before patients were exposed to systemic treatment [68]. These studies incorporated both ddPCR and non-ddPCR technologies, targeting mutations in *BRAF*, *NRAS*, *KIT* and *TERT* promoter. In more than 1000 patients tested with ddPCR methodology, detection of ctDNA at baseline was associated with a higher risk of disease progression, with HR = 2.10 (95% confidence interval [CI]: 1.71–2.59). It was also associated with higher risk of disease progression HR 2.15 (95% CI: 1.35–3.41) in 347 patients tested with non-ddPCR methodology. The prognostic value of ctDNA detection remains robust despite the type of treatment received, targeted or immunotherapy, for either detection method. The hazard ratio for death was 3.09 (95% CI 2.29–4.17) for immunotherapy-based treatment (*n* = 360) and HR 2.39 (95% CI 1.76–3.25) for targeted treatment (*n* = 666). The same meta-analysis confirmed that ctDNA positivity in metastatic melanoma is independently associated with higher disease staging, elevated LDH and visceral metastases, among other clinical characteristics that are historically linked with worse disease prognosis. Moreover, independent studies from McEvoy and Seremet and colleagues in a total of > 100 patients have demonstrated good correlation between ctDNA levels and 18F-fluoro-D-glucose positron emission tomography (FDG PET), an imaging tool that has become the standard of care in patients with advanced melanoma [69,70]. Therefore, ctDNA can be employed as companion test to inform disease prognostication alongside other widely used parameters.

***ctDNA in melanoma minimal residual disease.*** The powerful role of ctDNA in reflecting residual microscopic disease was initially showed in 2008, in patients with colorectal cancer. Diehl and colleagues demonstrated that patients with incomplete tumour resections had lower rate of ctDNA clearance or rising ctDNA fractions postoperatively compared to patients with complete resections, [71]. In melanoma, the detection rate of ctDNA after complete surgical resection can be up to 25% when using highly sensitive NGS-based assays [59,72,73]. Admittedly, low ctDNA quantity and sequencing artifacts can restrict the use of large sequencing panels in the adjuvant setting, and methods to reduce background error rate will be necessary, in order to maximise analytical specificity [74].

Multiple small-scale studies have demonstrated a significant correlation between the presence of peri-operative ctDNA detection and inferior survival outcomes in resected melanoma [73,75] in patients with known mutation variants. More recently, tumour-informed ctDNA panels that encompass wide range of patient-specific mutations have also demonstrated that peri-operative detection of patient-customized genomic variants are possible and predictive of patient outcomes [76,77]. One of the most robust reports on the predictive value of ctDNA in resectable melanoma was conducted and reported as part of the CheckMate 915 study translational analysis. Using a tumour-informed, patient-guided panel of 200 variants, Long and colleagues observed that in a cohort of 1127 patients with resected stage IIIB-D/IV melanoma, ctDNA detection at baseline can be an efficient predictor of recurrence-free survival for patients undergoing ICI treatment [78]. When ctDNA detection was combined with the level of tumour mutational burden (TMB) and a tumour-derived interferon signature (including HLA-DRA, CXCL9/10/11, GZMA, PRF1, CCR5, IFNG, IDO1 and STAT1), the predictive value of the combined score was enhanced [79]. In a real-world study of 30 patients who were followed up after resection, in the adjuvant setting, tracking of molecular disease with a tumour-informed ctDNA assay showed a sensitivity of 83% for distant relapses, with a specificity of 96% [80]. Importantly, ctDNA analysis allowed for an average lead time of 3 months over detection of disease recurrence with standard radiological assessment.

The realisation that ctDNA can give us great insight into the molecular tumour burden inspired further questions into how we can take advantage of this intricate information and how to best incorporate it into clinical decisions. In early stage colorectal cancer (stage II), the DYNAMIC study showed that foregoing adjuvant chemotherapy in patents with undetectable ctDNA post-operatively does not compromise recurrence free survival [81], and hence it can spare unwarranted chemotherapy toxicity. There are ongoing longitudinal studies to elucidate the exact role of ctDNA in detecting early molecular disease (NCT05736523) in early stage melanoma. Moreover, efforts to incorporate ctDNA in the stratification of patients and escalation/de-escalation of systemic adjuvant treatment are also underway. In “Tiragolumab Plus Atezolizumab Versus Atezolizumab in the Treatment of Stage II Melanoma Patients Who Are ctDNA-positive Following Resection (NCT05060003)” [82], adjuvant treatment was titrated based on the ctDNA positive detection following definitive surgery for stage II melanoma.

***ctDNA in advanced melanoma—molecular response to treatment.*** Molecular response and ctDNA kinetics refer to VAF dynamics during treatment, and it is worth mentioning that, besides tumour driver mutations, these can encompass the characterisation of neoantigens and/or chromosomal number aberrations [83,84,85]. Several studies and meta-analyses have attempted to highlight the robustness of ctDNA as a biomarker of treatment response, which can guide treatment duration or monitoring intensity in a pan-cancer fashion [59,62,68,86,87,88]. Across 16 tumour types, including melanoma, Zhang et al. confirmed that mean VAF pre-treatment with immune checkpoint inhibition was prognostic of overall survival [OS; stratified by the median; unadjusted HR, 0.58; 95% confidence interval (CI), 0.49–0.69; *p* < 0.0001]. Furthermore, on-treatment VAF dynamics correlated with overall response rate by radiological criteria [86]. This indicates that ctDNA is predictive of response to immunotherapy in the metastatic setting, and the authors concluded that a ratio-based molecular response metric can be used to guide clinical decisions. This ratio of on-treatment to baseline VAF demonstrated stronger association with RECIST response compared with on-treatment VAF alone (AUC = 0.82; 95% CI, 0.71–0.93 for the ratio and AUC = 0.73; 95% CI, 0.61–0.85 for on-treatment VAF). With the use of this score, patients could be stratified into “*molecular responders*”, and this molecular response could also predict future radiological response in patients who initially experienced radiologically stable disease, with a median lead time of 8 weeks. The meta-analysis from Zhang and colleagues included studies with either anti-PD1 or anti-PD1 plus anti-CTLA4 treatments [86].

In accordance with the above findings, in melanoma-specific studies, dynamic changes to quantified ctDNA demonstrate strong predictive value in estimating survival outcomes and response to immunotherapy in patients with metastatic disease [7,89,90,91]. With either immunotherapy, or targeted treatment, Varaljai and colleagues observed that a decrease in *BRAF* V600E ctDNA levels preceded radiologic detection of response in 80% of melanoma responders with an average lead-time window of 1.5 months (range, 0.023 to 3.45 months; *p* = 0.003). On the other hand, 86% of non-responders experienced an increase in ctDNA levels, which preceded radiologic progression, with an average lead-time window of 3.5 months (range, 0.23 to 18.86 months; *p* = 0.001) [92]. A similar trend was observed in the dynamics of *NRAS* and *TERT* promoter mutations. ctDNA decreases were more rapid with targeted treatment, which is in keeping with the symptomatic improvement that patients commonly experience when they start treatment with BRAF/MEK inhibitors in clinical practice. Interestingly, the same study demonstrated that *NRAS^Q61^* mutation can be detected with high-sensitivity ddPCR in the plasma of patients with tissue *BRAF* mutations, raising the hypothesis that *NRAS^Q61^* mutations might co-exist at low frequency in bulky tumours and suggests plasma ctDNA as a complementary genotyping tool too [92]. Nevertheless, anticipating disease response or progression is of paramount importance for the clinicians who, not infrequently, face discordance between clinical and radiological picture. Lee et al. observed that sensitivity of ctDNA for predicting pseudo-progression was 90% (95%CI, 68–99%) and specificity was 100% (95%CI, 60–100%) [8]. This indicates that ctDNA quantification is a valuable tool in distinguishing pseudo-progression from true progression on immune checkpoint treatment, an observation reported by others too [93] and in different cancer types [6,89]. Interestingly, ctDNA has also emerged as a biomarker during the novel treatment of tumour-infiltrating lymphocytes (TIL) in melanoma. Xi et al. reported the intriguing observation that patients with an early peak of ctDNA and subsequent clearance following TIL therapy experienced durable disease responses [94].

Overall, there is a multitude of studies confirming the validity of ctDNA as a biomarker of response to either targeted therapy or immunotherapy strategies [7,69,70,95,96,97,98,99,100,101,102,103,104,105,106]. A study that is strongly testing the ability of ctDNA measurement to aid clinical decision making in the metastatic setting is the CaCTUS trial (NCT03808441) [107]. In the CaCTUS study, patients with *BRAF* mutant melanoma on the intervention arm switch from targeted treatment to immunotherapy when there is evidence of molecular response, as defined by a decrease in mutant *BRAF* VAF of ≥80% by ddPCR. Results from this study will likely bolster ctDNA even more as a dynamic biomarker that can be incorporated into daily clinical practice of BRAF-mutated cutaneous melanoma. In a similar vein, the REPOSIT study attempted to investigate ctDNA mutational markers of resistance to targeted treatment with BRAF/MEK inhibition in unresectable stage IIIc/IV melanoma [108]. The study however was prematurely terminated due to slow accrual and the changing landscape of immunotherapy in advanced melanoma.

***ctDNA in non-cutaneous melanoma***—In uveal melanoma (UM), recurrent mutations affecting *GNA11*, *GNAQ*, *PLCβ4* or *CYSLTR2* are found in the majority of patients and, even if they are not determinants of prognosis, they can represent targetable mutations in the blood [37,109]. Using deep sequencing and hybridization capture to detect very low-frequency somatic alterations in exons and introns from 129 oncogenes (including *GNAQ*, *GNA11*, *SF3B1* and *EIF1AX*), Francis and colleagues observed a ctDNA detection rate of 29% during the perioperative period for primary UM [110]. This was supported by another study that reported a detection rate of 26% in patients with non-metastatic primary UM [111]. Interestingly, at follow-up, patients with ctDNA that became detectable or had an increasing VAF were significantly more likely to develop metastatic disease compared to patients with no detectable ctDNA or with decreasing VAF. In a bigger study of 135 patients with primary UM, ctDNA sensitivity and specificity of detecting metastases were found to be 80% and 96%, respectively [112], with an average lead time of 5.7 months (range of 2–10 months). To add to this, at the time of metastasis development, the presence of ctDNA seemed to be a strong predictor of overall survival, as showed in a prospective study of 179 patients [113]. An interesting and promising approach to the detection of ctDNA in UM patients with low burden disease was introduced by Wong et al. In a small study of 11 patients, the authors attempted to develop a customized multi-modal approach of ctDNA characterisation using genome, fragmentome and methylome analyses with targeted sequencing (TS), shallow whole-genome sequencing (sWGS) and cell-free methylated DNA immunoprecipitation sequencing (cfMeDIPseq) [114]. Interestingly, they observed that fragmentomic analysis of sWGS data was more effective at detecting ctDNA abundance in patients preceding disease relapse when compared to TS. As will be discussed further below, cell-free DNA fragmentomics can carry useful and complementary information that is agnostic genetic alteration agnostic.

In the context of established metastatic disease, Mariani showed that patients with hepatic metastases and detectable ctDNA have inferior survival outcomes compared to patients with undetectable ctDNA after liver metastasectomy (median recurrence-free survival: 5.5 vs. 12.2 months; HR = 2.23, 95% CI: 1.06–4.69, *p* = 0.04) [115]. In an exploratory analysis of a multicentre single-arm phase 2 study evaluating tebentafusp efficacy in previously treated metastatic UM, the level of ctDNA was measured in 127 patients with a uveal melanoma customised panel of common genomic aberrations, in which ctDNA was amplified using multiplex PCR and analysed with next-generation sequencing (performed by Natera Inc.) [116]. Both baseline levels and on-treatment reduction of ctDNA were strongly associated with overall survival. Patients with ctDNA clearance at 9 weeks after initiation of tebentafusp had longer survival probability (HR 0.08; 95% CI 0.01–0.56, *p* < 0.05), with an overall one-year survival rate of 100% compared to 52% in patients with increasing ctDNA. Given the observed disconnect between radiological response and survival outcomes in UM patients treated with tebentafusp [117], and given the intense and laborious nature of this weekly treatment, an urgent effort to incorporate markers of molecular response is needed. Measurement of ctDNA appears to offer a solution to both prognostication and prediction of response, but it deserves further prospective validation [118].

Ongoing current studies investigating the role of ctDNA and other minimally invasive biomarkers are summarized in Table 1.

## 5. Methylated ctDNA as a Biomarker and Novel Approaches

***DNA methylation.*** DNA hypermethylation at CpG islands and whole genome hypomethylation are both epigenetic hallmarks of melanoma [119,120,121]. A gradual gain of DNA hypermethylation status has been observed to occur in parallel with tumour aggressiveness. Characteristically, methylation of tumour suppressor genes’ promoter regions leads to their aberrant downregulation, which contributes to both oncogenesis and disease transformation/aggressiveness [122].

Recently, a lot of focus has been directed towards characterising ctDNA methylation status and the generation of methylation signatures that could function as disease biomarkers [123,124,125]. One of the candidate genes tested was tissue factor pathway inhibitor 2 (*TFPI2*), which was found to be methylated in the sera of patients with melanoma with the magnitude of methylation being more pronounced in the metastatic setting [126]. Hoon and colleagues examined the serum of patients with melanoma and discovered that the incidence of TSG hypermethylation increased during tumour progression and that specifically the hypermethylation of *MGMT*, *RASSF1A* and *DAPK* was significantly higher in metastatic tumours compared to primary melanoma tumours [127]. Moreover, others showed that a wider panel of genes can be interrogated via a ctDNA methylation analysis workflow using an amplicon-based NGS panel performed on bisulphite-treated DNA [56]. Diefenbach and colleagues confirmed the hypermethylation of seven genes (*GJB2*, *HOXA9*, *MEOX2*, *OLIG3*, *PON3*, *RASSF1* and *TFAP2B*) known to be hypermethylated in patients with metastatic melanoma but not healthy controls. Moreover, they showed that this serum signature of methylated tumour-related genes can act as a predictive marker of response to treatment with chemotherapy or immunotherapy and predict survival outcomes [128]. Targeted hypermethylated genes in this study were *RASSF1A* and *RAR-SS2.* The studies above are limited to locus-specific analyses of known genes and involved the methodology of sodium bisulphite modification, which is notoriously laborious and lengthy.

***Tissue-independent novel techniques.*** In an attempt to interrogate a larger part of the genome, Liu and colleagues developed an NGS-targeted methylation sequencing assay to measure the methylation status of more than 9000 CpG sites, selected according to TCGA data. This assay was used to predict tumour origin in a study including various tumour types, including melanoma. The methylation scores derived detected the presence of cancer in ~84% of the cases, and methylation-based signatures accurately classified the underlying cancer type in almost 79% of these [123]. Subsequently, further research revealed novel ways to dissect either large parts of or even the whole genome length, in order to discover signatures that could detect ctDNA with higher sensitivity and without the need for primary tissue. By testing the methylation status of >450,000 CpG islands of circulating free DNA (cfDNA), Moss and colleagues created a methylation atlas that can predict the presence of malignant DNA in circulating blood [129]. Additionally, the application of an immunoprecipitation-based protocol to sequence the entire cell-free DNA (cfMeDIPseq) detected large scale methylation patterns that are enriched for specific cancer types [130]. These novel techniques will undoubtedly be used in the near future for clinical validation in melanoma, where the challenges of primary tumour scarcity could be solved with the above tissue-independent assays. Moreover, additional epigenetic features of cfDNA, such as DNA fragment length are emerging as markers of underlying tumour biology and deserve further exploration in melanoma [131,132,133,134,135,136].

## 6. Limitations and Challenges with ctDNA

***ctDNA in melanoma CNS metastases.*** One of the main limitations of ctDNA application particularly relevant to melanoma is the poor sensitivity in the context of exclusively intracranial disease [68,69,101,104,137]. Lee et al. showed what was essentially 0% detectability rate in patients with brain metastases only and a poor concordance of ctDNA response to intracranial disease response in patients who had mixed intra- and extracranial tumour burden [137]. In the same vein, Seremet and colleagues found that melanoma progression in the brain does not correlate with ctDNA detection, and similar observations have taken place in other tumour-type contexts too, making this a more universal phenomenon [70,138,139,140]. The known impact of the blood–brain barrier in hindering ctDNA from entering systemic circulation is a hypothetical culprit. A potential avenue for this challenge could be ctDNA measurement in cerebrospinal fluid (CSF), which has demonstrated improved sensitivity [141,142,143]. Due to the invasive nature of CSF sampling, there has been no large-scale study validating the wider applicability of CSF ctDNA measurement; nevertheless, case studies have showed that it reflects the disease dynamics in melanoma [144,145]. The low detectability of ctDNA in patients with brain metastatic disease could potentially be circumvented with the use of cfMeDIPseq. This novel technique that characterises methylome signatures in the blood was found to accurately discriminate between brain tumours based on their cell of origin, and based on their primary or metastatic nature, in a tissue-independent manner [146], and could potentially replace the use conventional ctDNA characterisation in patients with intracranial metastases from melanoma.

***Technical challenges***. In early-stage melanoma, where ctDNA shedding might be limited and specifically for the triple wild-type subtype, the need to optimise and validate DNA-based biomarkers with higher sensitivity is paramount [147]. Interpretation of results need to consider that in early disease, the ctDNA amount might not be enough to detect single nucleotide mutation that have been detected in tumour tissue [148]. In small primary melanoma tumours, there might not be enough tissue to implement a wide molecular genomic panel for full characterisation to begin with. Tissue-independent and agnostic methodologies, such as the methylation/fragmentomic signatures already mentioned, could provide some avenues for development. Moreover, exploring the tumour fraction profile of ctDNA, rather than targeting specific gene loci, also presents an interesting alternative [88]. Contamination of the results by DNA fragments derived from clonal haematopoiesis of indeterminate potential (CHIP) or non-neoplastic haematopoietic stem cells can lead to false-positive ctDNA results. This can be mitigated by advanced bioinformatic analysis or by filtering out mutations by sequencing matched tumour and leucocytes derived by the same sample [132,149]. Lastly, despite the myriad of techniques and technologies developed over the last few years for the detection of ctDNA, standardization regarding the wider use of a particular assay, streamlined analysis and even sampling timing are still not well defined. Recent intriguing evidence suggesting that intravasation of circulating tumour cells is circadian rhythm-dependent could signify that even the exact time of ctDNA sampling could affect the final result [150]. An overview of the potential application of ctDNA and other minimally invasive biomarkers are depicted in Figure 2.

## 7. Circulating Melanoma Cells

***Circulating tumour cells—the technique.*** Tumour progression results from a complex combination of genomic alterations that ultimately confer neoplastic cells with the capability to exit the primary site and migrate to distant tissues, primarily through haematogenous spread. During this process, primary tumour cells undergo genomic and structural alterations known as epithelial-to-mesenchymal transition, enabling them to enter the circulatory system as circulating tumour cells (CTCs). CTCs travel in the blood stream until permeation of the endothelium allows them to infiltrate distant tissues [151]. Minimally invasive blood sample collection, combined with novel isolation and purification techniques, presents CTCs as an appealing tool of liquid biopsy approaches in oncology. The methodologies for CTC acquisition have been evolving to ensure the capture of viable cells at the appropriate concentration, enabling the performance of downstream analyses across all levels of genomic, transcriptomic, proteomic and epigenomic readouts [152]. Among various techniques is the antibody-coated magnetic bead methodology directed towards surface proteins such as EpCAM, pan-CK and CD45. Additional approaches target specific physical features of CTCs, such as density-based separation [153]. Recent developments in techniques are grounded in the combination of physical and biological CTC features, such as fluid dynamics and membrane–cell superficial markers. Novel CTC chip devices exhibit the capability of efficiently capturing CTCs from whole blood samples within silicon micro-posts, integrating parameters such as cell flow velocity and cell attachment attributes within chambers housing anti-epithelial cell adhesion molecule (EpCAM) antibodies [152].

CTC isolation in the blood can predate the conventional, radiological imaging-based detection of metastatic disease, allowing for earlier detection of advanced status [154]. In the clinic, this could guide treatment decisions into preventing symptoms development and clinical deterioration [155]. Notably, simultaneous assessment of these circulating components in blood samples may offer a more comprehensive reflection of the inherent tumour heterogeneity than compare to the conventional evaluation of a single-site tissue biopsy, with *KRAS* mutational profile representing an example [156,157]. CTC levels also hold predictive value, as demonstrated in a prospective non-inferiority phase III trial that utilized a CTC threshold (≥5 CTCs/7.5 mL) as a stratification factor between adjuvant chemotherapy or endocrine therapy alone in localized hormone receptor-positive *HER2*-negative breast cancer patients [158]. This study successfully reached its non-inferiority progression-free survival endpoint, showing that CTC count is a superior biomarker to the conventional stratification approaches, such as clinical and pathological tumour features [158]. Prospectively, the genomic analysis of prostate CTCs for the androgen receptor splice variant, AR-V7, could serve as a predictive tool for assessing the clinical benefits of enzalutamide or abiraterone in terms of prostatic surface antigen (PSA) and imaging responses, within the context of metastatic castration-resistant disease. Patients with CTCs expressing the AR-V7 variant experienced inferior outcomes in terms of PSA response rates, PSA progression-free survival, clinical and radiographic progression-free survival, and overall survival in comparison to those lacking the expression of this variant [159].

***Circulating tumour cells in melanoma.*** Broadly, evidence suggests that melanoma cells spreading from primary tumours follow the same process described previously, supporting the presence of melanoma CTCs (mCTCs) in blood stream at all disease stages [160]. Although mCTCs lack the expression of membrane epithelial markers usually necessary for cellular isolation (e.g., EpCAM), they express other specific markers such as melanoma cell adhesion molecule (MCAM), paired box gene 3 d isoform (PAX3d), microphthalmia-associated transcription factor m isoform (MTIFm) and transforming growth factor-beta 2 isoform (TGF-β2) that can be used for diagnostic purposes, as well as prognostication in cases of advanced disease [160,161].

The potential role of mCTCs in uveal melanoma diagnosis was investigated in a small prospective study involving eight treatment-naïve patients and four healthy individuals with choroidal nevi. The detection of mCTCs was carried out via a semi-automatic CellSearch system (Veridex, Warren, NJ, USA) utilizing magnetic beads coated with anti-CD146 and anti-MEL (high-molecular-weight melanoma-associated antigen) from peripheral blood samples. In this study, a minimum of one mCTC/7.5 mL of blood was identified in four out of eight patients with uveal melanoma, while no mCTCs were detected in the control group. Notably, the sole patient who presented with three mCTCs/7.5 mL also exhibited an extra scleral extension of the primary tumour [162].

The assessment of minimal residual disease by identifying or quantifying mCTCs in the bloodstream may contribute to a more precise evaluation of the risk of recurrence, which, in cutaneous melanoma, has traditionally relied heavily on pathologic tumour features such as Breslow thickness, ulceration, lymph node involvement and others, particularly in the early stages of the disease. In an exploratory analysis of the phase III trial EORTC 18991, which compared adjuvant pegylated interferon-alpha-2b with surveillance in patients with resected stage III melanoma, the presence of mCTCs was verified using RT-PCR for tyrosinase and Mart-1/Melan-A transcripts. Patients who had detectable mCTCs during the follow-up period experienced a significantly increased risk of distant relapse (hazard ratio of 2.23; 95% confidence interval, 1.40–3.55; *p* < 0.001), regardless of whether they received adjuvant treatment or not [163]. This underscores the significance of mCTCs as a prognostic factor in this context. Similar findings were observed in the metastatic setting in a prospective study conducted by Khoja et al., which explored mCTC quantification using the previously described CellSearch system in 101 patients in the early era of clinical immunotherapy. In this study, a cutoff of ≥2 CTCs/7.5 mL at baseline was associated with significantly poorer overall survival compared to those with <2 CTCs/7.5 mL (median overall survival of 7.2 months vs. 2.6 months, respectively; hazard ratio of 0.43, 95% confidence interval, 0.22–0.81; *p* = 0.009). Even after adjusting for other factors such as *BRAF* status, treatment type and time to diagnosis of metastatic disease, the cutoff of ≥2 CTCs/7.5 mL remained a significant poor prognostic factor (*p* = 0.005) [164].

Using parameters such as receptor expression and cellular physical characteristics can enable the extraction of valuable information from mCTCs. In a cohort of 43 patients with metastatic melanoma, positive mCTC detection by a combination of droplet digital PCR (ddPCR), RT-PCRand immunocytochemistry had a statistically significant correlation with inferior overall survival (HR for death of 7.8) when compared to patients without mCTCs prior to systemic treatment with immunotherapy or targeted therapy. Furthermore, high transcriptomic mCTC scores (cut-off of more than 100 transcripts per ml of blood based on ddPCR analyses) were also associated with inferior survival in the same study [165]. In addition, a comprehensive examination involving mRNA analysis (employing qRT-PCR for *MAGEA3*, *MLANA*, *B4GALNT1*, *PAX3* and *DCT*), ddPCR for *BRAF* V600R mutation, as well as lactate dehydrogenase levels, in stage III/IV mCTC patients undergoing immune checkpoint inhibitor therapy, demonstrated the capacity to classify patients into two distinct and cohesive risk categories utilizing a decision-tree methodology. Those identified as high-risk exhibited notably inferior disease-free and overall survival outcomes within this context [166]. When researchers specifically looked into response to anti-PD-1 inhibition, the expression of PD-L1 in circulating tumour cells (CTCs) holds potential as a predictive indicator of pembrolizumab’s efficacy within the metastatic context. Khattak et al. have reported that patients with positive PD-L1 status in mCTCs prior to treatment, as detected by flow cytometry, exhibit a progression-free survival of 26.6 months, in contrast to 5.5 months observed in PD-L1-negative mCTC cases. This difference was validated through multivariate analysis for the aforementioned endpoint [167].

The current absence of high-sensitivity standardized techniques for membrane markers associated to mCTCs, necessitates a significant volume of blood to facilitate a satisfactory isolation process. This presents a significant restricting factor in the development of CTCs’ clinical utility. Further challenges, such as the notably low success rates observed in ex vivo CTC cultures, also present considerable obstacles to their clinical translation [168]. On-going current studies investigating the role of mCTCs and other minimally invasive biomarkers are summarized in Table 1.

## 8. Melanoma Derived Extracellular Vesicles and Coding/Non-Coding RNA

***Extracellular vesicles.*** Tumour cells are capable of intercellular communication via the secretion and packaging of intracellular signalling components (e.g., RNA, lipids, proteinsand DNA) into bilayer phospholipid membrane vesicles sized between 30 nm to 10 μm [169,170]. These extracellular vesicles (EV) can influence the local microenvironment or be exported to distant recipient tissues, ultimately fostering an environment that is conducive to tumour infiltration [171,172]. EVs are classified according to their size in exosomes (30 nm–150 nm) or shed microvesicles (50 nm–1.300 nm), with each exhibiting distinct biogenetic features [173]. The formation of exosomes is associated with the recycling processes of membrane multivesicular elements, as exemplified by endocytosis, wherein intracellular vesicles are generated through the invagination of the plasma membrane, subsequently undergoing loading and processing of the components of these vesicles for subsequent release into the extracellular environment [174]. On the other hand, shed microvesicles originate from the budding of cytoplasmic material surrounded by the plasma membrane through reorganization and contraction of the actin–myosin cytoskeleton near the cell periphery [175].

EV-mediated intercellular physiological communications are subverted by cancer cells in many steps involved in tumorigenesis. EVs transporting cargo could be selectively enriched with various types of coding and non-coding RNA capable of influencing transcript programs locally or in distant tissues [171]. Recent analyses have revealed that mRNA, miRNA and long non-coding RNA (lncRNA) represent the predominant nucleic acids within EVs. These can be assessed through methodologies such as quantitative polymerase chain reaction (qPCR), next-generation sequencing or microarray techniques [176]. Serum breast cancer associated exosomes contain-miRNAs that are capable of initiating tumorigenesis of normal epithelial cells trough modification of their transcriptomic program [177]. Similarly, patients’ culture-derived glioblastoma cells can secrete exosomes enriched in mRNA, miRNA and proteins capable of incorporation by brain endothelial cells and subsequent stimulation of angiogenesis. Moreover, *EGFR*vIII mRNA EV-cargo, a truncated form of epidermal growth factor receptor specifically expressed by glioblastoma cells, can be detected in patients’ serum, transferred to and merged with other glioma cancer cells lacking this marker and leading to upregulation of their oncogenic activity [178,179].

Interestingly, tumour-derived exosomes can exhibit tropism towards certain distant tissues owing to their encapsulation of tumour-specific adhesion molecules, such as integrins. Within these tissues, the cargo carried by EVs can instigate changes in the extracellular matrix, vasculature and intratumoural immune microenvironment. These modifications can bolster a microenvironment conducive to tumour colonization. The presence of these adhesion molecules could elucidate the predilection for metastatic localization in organs like the lungs and liver [180]. Aligned with the strong interaction between melanoma and the immune system, exosomes derived from the plasma of patients with melanoma exhibit a higher enrichment of immunosuppressive proteins when compared to healthy controls. These exosomes demonstrate the ability to hinder the activity of both CD8^+^ T cells and NK cells in vitro [181].

EV cargo is cell-specific and mirrors factors such as cellular metabolism and genomic profile [182,183]. Moreover, in vitro experiments have demonstrated that tumour cells exhibit increased amounts of EV compared to their normal tissue counterparts [184]. Interestingly, some tumour-associated EVs could be found in higher concentrations in different body fluids of cancer patients compared to healthy controls [185]. Exploring these characteristics allowed for the development of methods of measuring the EV-related components which could be used in clinical practice such as the evaluation of RNA contained-exosomes in urine samples for the diagnosis of high-grade prostate cancer [186,187,188]. These isolation techniques integrate physical EV features with cellular-origin specific markers to enhance the efficiency and purity of the isolation process [189]. For example, accurate characterization of surface EV proteins is essential to permit the production of specific monoclonal antibodies that could segregate EV subtypes for widespread differently clinical uses [171].

***EVs in melanoma.*** Small-scale studies have revealed the clinical potential of melanoma-specific EV cargo measurement, as biomarkers for both early and advanced disease stages [190,191]. The exosomal miRNA plasma profile could potentially aid in the diagnosis of melanoma with a substantial degree of accuracy, as evidenced by the diminished levels of EV-miR-1180-3p observed in patients in comparison to healthy controls (AUC for the ROC curve of 0.729) [192]. Mechanistically, miR-1180-3p expression negatively regulates malignant cell traits, such as proliferation, invasion and migration, in an in vitro melanoma cell model, possibly through post-transcriptional inhibition of the melanoma related-gene *ST3GAL4* expression [192]. In addition, elevated levels of serum melanoma exosome-associated proteins, namely S100 and MIA, demonstrated sensitivity in differentiating patients with metastatic disease against both disease-free and healthy individuals. Furthermore, heightened levels of serum MIA-associated exosomes proved to be a significant predictor of poorer survival outcomes [193].

In vitro, uveal melanoma (UM) cells demonstrated a greater propensity for EV shedding in comparison to normal choroidal melanocytes. This phenomenon extends to the clinical setting, wherein the mean concentration of EVs positive for melanoma markers such as CD63 and TSG101 was significantly higher in the vitreous fluid, aqueous fluid and plasma of UM patients when compared to a control group devoid of cancer [194]. Notably, it is of interest that the cargo content of these EVs remained consistent regardless of the origin of the fluid samples in UM patients, thereby underscoring their potential utility as a tool for disease monitoring [194]. In accordance with these findings, Wróblewska et al. demonstrated a distinct EV-associated miRNA profile in exosomes isolated from the bloodstream of UM patients when compared to that of healthy individuals. Employing a real-time qPCR approach, the authors observed a significantly elevated level of miRNAs associated with pivotal processes related to malignant acquisition traits in affected individuals [195]. Notably, the following miRNAs displayed heightened expression levels: hsa-miR-191-5p, hsa-miR-223-3p, hsa-miR-139-5p, hsa-miR-10b-5p, hsa-miR-483-5p, hsa-miR-203a and hsa-miR-122-5. Among these, hsa-miR-191-5p, hsa-miR-223-3p, hsa-miR-483-5p and hsa-miR-203a exhibited the most pronounced sensitivity for diagnosing UM, with an accuracy exceeding 80% for each. Moreover, hsa-miR-191-5p and hsa-miR-144-5p showed potential for discriminating between localized and metastatic diseases, with the former displaying lower and the latter higher expression levels. However, their accuracy in doing so was less robust, as indicated by area under the ROC curve values of 0.71 and 0.86, respectively [195]. Reinforcing the potential role of miRNAs as prognostic tools, Sun et al. developed an integrated risk signature for overall survival in UM-affected individuals. This signature was developed using the five highest-performing serum miRNA expression profiles (hsa-miR-513a-5p, miR-506-3p, miR-508-3p, miR-140-3pand miR-103a-2-5p), which were combined with clinical information and their respective target genes using Weighted Correlation Network Analysis (WGCNA). This risk signature consistently demonstrated the ability to differentiate between low and high-risk groups, as evidenced by Kaplan–Meier analysis (*p* < 0.001), ROC curve analysis (accuracy > 0.9) and its status as an independent prognostic factor according to multiple Cox regression analysis (hazard ratio for death of 3.799; 95% CI: 1.903–7.583; *p* < 0.001) [196].

Notably, these vesicular markers also hold promise for treatment monitoring. A correlation was observed between reduced plasma levels of four specific EV-melanoma membrane-bound proteins (*MCSP*, *MCAM*, *ERBB3* and *LNGFR*) and response to treatment involving combination BRAF/MEK inhibitors in patients with metastatic disease harbouring the *BRAF* V600E mutation [197]. When examining soluble and exosomal PD-L1 in patients with melanoma, Cordonnier and colleagues highlighted the fact that exosomal PD-L1 was higher in concentration compared to soluble PD-L1 and that the exosomal form retained immunosuppressive properties. Intriguingly, diminished EV-PD-L1 plasma levels were prospectively linked to favourable responses to systemic treatment (specifically anti-PD1 inhibitors), in the context of metastatic disease, demonstrating strong accuracy (AUC of 0.867 for ROC curve) [198]. This raises hope that EV-PD-L1 could represent a reliable and dynamic biomarker of response to immunotherapy. Another potential avenue for use of EVs is treatment monitoring. Shi et al. showcased a robust concordance between plasma EVs transcriptomic profiles and those derived from bulk tumour biopsies. The assessment of EV transcriptomic profiles had the potential to delineate genomic signatures for predicting resistance to immune checkpoint inhibitors through dynamic evaluations before and during treatment, albeit with a marginally lower accuracy compared to that achieved through bulk tumour biopsies. Furthermore, the precision of these EV-genomic signatures can be enhanced through the application of a deconvolution model, allowing for a more accurate differentiation between tumoural EV-transcripts and those originating from non-tumoural sources [199].

Despite these promising prospects, employing EVs as clinical decision tools demands resolution of pivotal issues. These include the standardization of vesicle purification methodologies, the refinement of EV tumour-specific biomarkers not expressed by normal cells and their validation through large prospective studies (Table 1) [187].

***Circulating free MicroRNA (miRNA) and long non-coding RNA in melanoma.*** Preliminary analysis has revealed abnormal expression profiles of miRNA in various tissue samples, including blood, from cancer patients compared to healthy individuals [200]. These findings have prompted further studies investigating the role of miRNA in carcinogenesis, as well as its potential utility as a diagnostic, prognostic and predictive tool in the field of melanoma [201]. In preclinical models, miR-221 plays a pro-oncogenic role in melanoma cells by downregulating the expression of the cellular cycle modulator p27Kip1/CDKN1B. This finding has been translated to clinical relevance as indicated by a positive association between serum miR-221 levels and more advanced disease stages as well as increased tumour thickness. Additionally, the concentration of miR-221 in serum may prove useful for disease monitoring following primary tumour resection [202]. A multicentre study, encompassing centres in Australia and Germany, utilizing a refined panel of 7 melanoma-related miRNAs (MELmir-7) measured in serum, has demonstrated a sensitivity of 93% and specificity of 82% for the detection of clinically occult metastatic disease. Additionally, it exhibited superior performance in detecting melanoma recurrences compared to serum measurements of LDH and S100B [203]. Aligned with these findings, Huber et al. illustrated a statistically significant elevation in serum levels of miRNAs associated with myeloid-derived suppressor cells (miR-146a, let-7e, miR-125aand miR-145b) in patients diagnosed with metastatic melanoma when compared to healthy controls. Moreover, the study demonstrated that elevated baseline serum levels of these miRNAs were correlated with diminished overall and progression-free survival in patients exposed to a combination of immunotherapy and targeted therapy in the metastatic setting [204].

Recent studies have brought to light the role of non-coding RNAs (ncRNAs), distinct from EV-associated miRNAs, in melanoma carcinogenesis, underscoring their potential as diagnostic tools and innovative therapeutic targets [205]. Previous analyses have unveiled heightened levels of *SPRIGHTLY*, a lncRNA, engaged in intracellular lipid regulation, within the plasma of melanoma patients as measured by qPCR, in contrast to healthy donors [206]. Furthermore, a cutoff of a 2.64-fold difference in cancerous/noncancerous *SPRIGHTLY* levels has been identified as a prognostic tool, achieving an ROC accuracy of 0.813 (*p* < 0.001) and correlating with diminished survival among high-expression melanoma patients, compared to their low-expression counterparts (38 vs. 51 months; *p* < 0.001). In concordance with these findings, RNA interference (RNAi) targeting *SPRIGHTLY* has been demonstrated to impede malignant behaviour, including migration and invasion and to diminish the viability of melanoma cells in vitro [207]. A pan-cancer RNA sequencing analysis, encompassing over 10,000 tumour specimens from the TCGA database, revealed a significant upregulation of Survival Associated Mitochondrial Melanoma Specific Oncogenic lncRNA (*SAMMSON*) in cutaneous and uveal melanoma (UM) samples compared to other tumour types. In UM patients, *SAMMSON* expression was also found to be associated with staging of metastatic disease. Biologically, *SAMMSON* expression appears to play a crucial role in melanoma cell viability, as demonstrated by its knockdown via antisense oligonucleotide (ASO), which resulted in apoptosis in various UM and cutaneous melanoma cell lines. Other analyses also revealed similar insights into circular RNAs (circRNAs), a specific type of ncRNA, in melanoma development. This was demonstrated by a distinct circRNA expression profile in tumour tissue samples when compared to adjacent normal tissue in six patients with lymph node-positive oral mucosal melanoma, as assessed by microarray analysis. The authors identified 58 upregulated and 32 downregulated circRNAs associated with melanoma samples, which were subsequently validated by qRT-PCR for the five circRNAs (upregulated: hsa_circ_0005320, hsa_circ_0067531, hsa_circ_0008042; downregulated: hsa_circ_0000869 and hsa_circ_0000853) among the top 10 altered ones, as compared with matched normal tissues. Additionally, bioinformatic analysis indicated that these three highly expressed circRNAs may be associated with tumourigenesis and metastatic facilitation by targeting specific microRNAs [208]. Similarly, Bian and colleagues demonstrated a high expression of hsa_circ_0025039 circRNA using a microarray approach, which was subsequently validated by qRT-PCR in melanoma tissue samples when compared to matched adjacent normal tissue samples [209]. The authors confirmed this expression profile in tumour samples from 43 patients with melanoma through qRT-PCR, comparing it to healthy skin samples. The authors also showed a negative prognostic value in patients with high expression compared to those with low expression (*p* < 0.05). Corroborating these findings, silencing hsa_circ_0025039 with siRNA was found to inhibit proliferation, impair cell invasionand reduce glucose consumption by melanoma cells in vitro. On-going current studies investigating the role of EVs and other minimally invasive biomarkers are summarized in Table 1.

## 9. Intestinal Microbiome as a Biomarker in Melanoma

Intestinal microbiota and their associated genome (microbiome) can modulate metabolism, antitumour activity and the safety of a broad variety of anticancer treatments, including immunotherapy [210,211,212,213,214]. We can infer the composition of the gut mucosal microbiome by examining the metagenomic landscape of the stool microbiome and this can give us invaluable information on how many different bacterial taxa are present in the host’s intestine i.e., on the taxonomic diversity. In a case control study, the gut microbiome diversity and composition were different between healthy controls, patients with early stage (stage I/II) and patients with late-stage melanoma (stage III/IV) [215]. Characteristically, α-diversity diminished in later stage melanoma.

An association between higher microbiota compositional diversity and superior benefit from anti-PD-1/PD-L1 immune checkpoint inhibition has been reported for several cancer types, including melanoma [216]. The abundance of specific bacteria populations such as Bifidobacterium and Ruminococcaceae has been detected in the gut microbiota of patients deriving benefit from immunotherapy [216,217]. On the other hand, the microbiome composition of non-responders is enriched for taxa such as Bacteroides and Clostridium species. Unsurprisingly, it has also emerged that antibiotic-induced gut microbiota dysbiosis adversely affects clinical outcomes during immune checkpoint inhibition [218,219,220,221,222]. So far, the mechanisms underlying the relationship between microbiota composition and response to immune checkpoint inhibitors suggest that microbiota can induce and activate multiple effector immune cells, such as NK, dendritic, CD4^+^ and CD8^+^ T cells, through interfering with the generation of several metabolites such as short-chain fatty acids, inosine and bile acids [223,224,225,226,227,228,229]. Furthermore, cross-reactivity between cancer and microbiota antigens can augment cancer immunogenicity and hence, promote immunotherapy efficacy [230,231]. Manipulation of the gut microbiome with faecal microbial transplantation can not only reverse the microbiome dysbiosis but also rescue anti-PD-1 resistance in patients with metastatic melanoma [224,232,233].

In addition to the role of gut microbiome in ICI responsiveness [217,233,234], imbalances in the composition or function of gut microbes has also been implicated in ICI toxicity, specifically in patients with melanoma [233]. Species belonging to Bacteroidetes, Clostridia, and Proteobacteria phyla have been linked to increased incidence and severity of immune-related adverse events (irAEs) [214,235,236]. Out of these events, ICI-related colitis has been most closely linked with the presence of members of the Firmicutes phyla (*F. prausnitzi*, *G. formicilis*, *Ruminococcaceae* spp.), during anti-CTLA-4 monotherapy and B. intestinalis (phylum Bacteroidetes) during combination CTLA-4/PD-1 immunotherapy [214]. Collectively, a microbial signature dominated by Streptococcus spp. is accompanied by worse clinical outcomes and a high frequency of distinct irAEs, such as immune-mediated arthropathy [233]. Intriguingly, a strong sequence and conformational homology exists between tumour-associated antigens and microbiome-derived peptides from species of the Firmicutes and Bacteroidetes phyla [237], suggesting molecular mimicry between tumour and microbiome epitopes as a cross-reactivity triggering event.

Notwithstanding all the interesting observations made in individual studies above, there is a notable lack of concordance and a common response-related microbial signature, when multiple studies are interrogated. Several factors may of course adversely influence the interpretation of data including small sample sizes, arbitrary definitions of clinical responders and non-responders and variations in bioinformatic analytic pipelines [238]. Importantly, there is widespread challenge in controlling for the confounding factors known to influence gut microbiome composition, such as medication intake, diet, geography and ethnicity [239,240]. In summary, given the complex and vast microcosmos of the gut microbiome, it might be significantly challenging to compile a unique and consistent compositional signature that would efficiently act as a biomarker of response or toxicity. It is conceivable that we would have to redirect our focus from compositional difference to differences in the overall functional capacity of an entire microbial community and discover a direct and quantifiable readout of this function, that could act as a biomarker.

## 10. Conclusions

For years, the oncology research community was entirely focused on the tumour microenvironment and the influence it exerts on its adjacent tissue. More recently however, it has become apparent that a lot of information can be extracted from circulating tumour products and their dynamics for both prognostication and treatment outcome prediction (Table 2). The initial excitement about circulating tumour cells seemed to have now been transferred to circulating tumour DNA, which is currently the most well-developed, minimally invasive biomarker. Tremendous effort has been directed towards its optimization and streamlining; and further on-going research is aiming at extracting all the possible information from circulating nucleic acid products, with the least possible nucleic acid volume required. Despite this, there are still on-going challenged pertaining to the complexity and cost of ctDNA and other minimally invasive biomarkers, indicating that their incorporation into clinical practice will require extensive work and validation (Figure 2). In the meantime, additional tumour products, such as extracellular vesicles and micro-RNAs can add further characterization and potentially work synergistically with ctDNA in create a complete molecular tumour profile. Last, but not least, the intestinal mucosal surface and its strong impact on modulating systemic immunity can also be exploited, with information residing into non-invasive stool samples. In keeping with biomarker development, standardization, harmonization, and cross validation will establish new discoveries into clinical practice. Serial monitoring of biomarkers in patients receiving treatment will inform escalation and de-escalation strategies. To achieve this, however, the incorporation of these biomarkers in large scale, national and international collaborative clinical studies, will be necessary.

## Figures and Tables

**Figure 1 cells-13-00019-f001:**
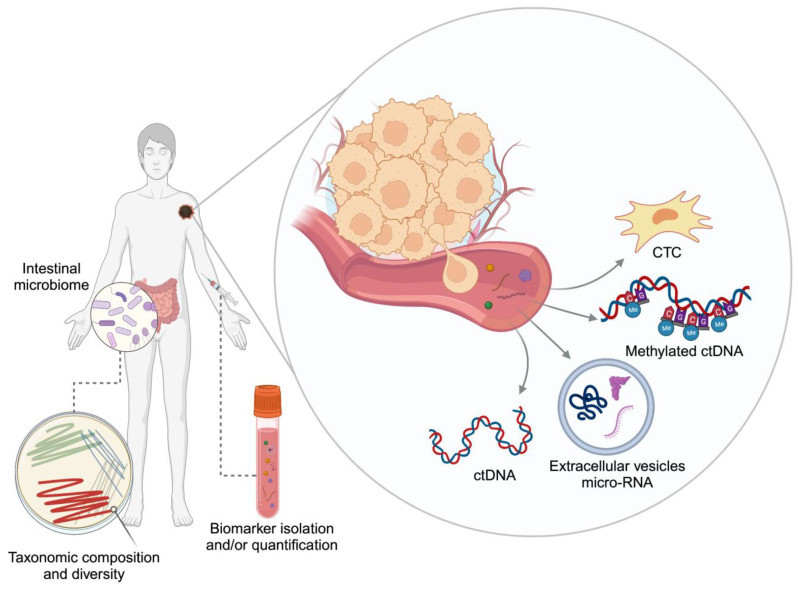
Minimally invasive biomarkers. CTC: circulating tumour cell, ctDNA: circulating tumour DNA. Created with BioRender.com.

**Figure 2 cells-13-00019-f002:**
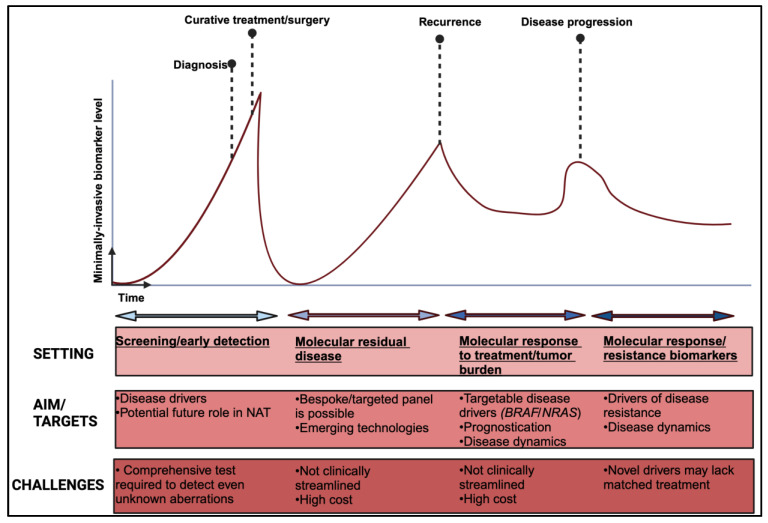
Overview of non-invasive biomarkers applications in melanoma. NAT: neo-adjuvant treatment. Created with BioRender.com.

**Table 1 cells-13-00019-t001:** Melanoma studies registered on clinicaltrials.org involving liquid biopsy methodologies, as of 1 October 2023.

Biomarker	Design	Clinical Scenario	Number of Participants	Primary Outcome	Study Status	Reference
**ctDNA,** **Exosomes**	Observation prospective	*BRAF* mutant melanoma patients	12	Percentage correlation between circulating tumour DNA and metastatic sites	Completed	https://clinicaltrials.gov/study/NCT02251314 (accessed on 1 October 2023)
**ctDNA**	Phase II trial	Unresectable stage IIIc/IV *BRAF* V600 mutant melanoma patients under treatment with vemurafenib plus cobimetinib	78	Treatment response	Terminated	https://clinicaltrials.gov/study/NCT02414750 (accessed on 1 October 2023)
**CTCs (isolated tumour cells from malignant fluids, core needle biopsies, fine needle aspirates or resections)**	Observation prospective	Adult patients diagnosed with any carcinoma undergoing treatment	200 (estimated)	Best overall Response and progression free survival	recruiting	https://clinicaltrials.gov/study/NCT05461430 (accessed on 1 October 2023)
**ctDNA**	Observation prospective	Early stage or locally advanced tumours that are planned for or have undergone curative treatment	500 (estimated)	Determine minimal residual disease	Recruiting	https://clinicaltrials.gov/study/NCT05196087 (accessed on 1 October 2023)
**CTCs**	Observation prospective	Recurrent or metastatic head and neck squamous cell carcinoma, non-small cell lung cancer, or metastatic melanoma which are going to receive checkpoint inhibitors	155 (estimated)	Clinical performance of PD-L1 kit in CTCs of peripheral blood and tumour tissue samples	Active not recruiting	https://clinicaltrials.gov/study/NCT04490564 (accessed on 1 October 2023)
**miRNA/ncRNA**	Observation prospective	Melanoma patients	300 (estimated)	Integration between molecular diagnostic and pathological staging parameters, imaging non-invasive instrumental diagnostic, dermatologic clinical diagnostic and complement to surgery	Recruiting	https://clinicaltrials.gov/study/NCT05906277 (accessed on 1 October 2023)
**Circulating cell-free nucleic acids, cfDNA, cfRNA**	Observation prospective	Patients with either histological confirmation of a solid tumour or haematological malignancy, or patients identified as high-risk for cancer (based on identified aberration in cancer predisposition gene or on hormonal and/or family history without known aberration).	2500 (estimated)	Collection and annotation of biospecimens	Recruiting	https://clinicaltrials.gov/study/NCT03702309 (accessed on 1 October 2023)
**ctDNA**	Observation prospective	Early stage solid tumours that have undergone definitive treatments	1000 (estimated)	Distant recurrence free interval	Recruiting	https://clinicaltrials.gov/study/NCT05059444 (accessed on 1 October 2023)
**Tumour circulating nucleic acids and proteins**	Observation prospective	*BRAF* V600E melanoma patients under adjuvant treatment	50 (estimated)	Multiplexed detection and quantification of protein and nucleic acid analytes with sensitivity at single-molecular level	Recruiting	https://clinicaltrials.gov/study/NCT05940311 (accessed on 1 October 2023)
**Exosomes**	Prospective single group assignment	Unresectable stage IIIc/IV *BRAF* V600 mutant melanoma patients who are considered for BRAF inhibitor treatment	15	Measure of the number of exosomes (µg of proteins or particles)/mL in peripheral blood by differential ultracentrifugation before and after treatment	Unknown	https://clinicaltrials.gov/study/NCT02310451 (accessed on 1 October 2023)
**Exosomes**	Observation prospective	Melanoma patients	150 (estimated)	Quantification of circulating exosomes	Active, not recruiting	https://clinicaltrials.gov/study/NCT05744076 (accessed on 1 October 2023)
**CTCs**	Prospective single group assignment	Patients with advanced melanoma stage IIIC (unresectable) or stage IV	30	Determination the effect of treatment on the number of circulating melanoma cells in patients with metastatic melanoma	Completed	https://clinicaltrials.gov/study/NCT01573494 (accessed on 1 October 2023)
**CTCs**	Observation prospective	Advanced melanoma patients	73	To compare results for the detection of circulating melanoma cells (CMC) using CellSearch versus EPISPOT (EPithelial ImmunoSPOT) techniques between a group of patients with metastatic melanoma and a group of hospitalized control patients	Completed	https://clinicaltrials.gov/study/NCT01573494 (accessed on 1 October 2023)
**ctDNA**	Phase II	Advanced *BRAF* V600E/K/R mutated melanoma stage IIIC (unresectable) or stage IV	21	To determine whether switching from targeted therapy to immunotherapy based on a decrease in levels of circulating tumour DNA in the blood will improve the outcome in melanoma patients.	Recruiting	https://clinicaltrials.gov/study/NCT01776905 (accessed on 1 October 2023)
**Exosomes, CTCs**	Observation prospective	*BRAF* mutant melanoma patients	12	Percentage correlation between circulating tumour DNA and metastatic sites	Completed	https://clinicaltrials.gov/study/NCT02251314 (accessed on 1 October 2023)
**ctDNA**	Prospective single group assignment	Locally advanced, operable melanoma treated with immunotherapy or anti-*BRAF* and anti-*MEK* targeted therapies (stage IIIb, IIIc) or exclusive immunotherapy (stage IV) in an adjuvant situation.	165	Studying the tumour molecular abnormalities resulting from circulating tumour DNA (ctDNA) to predict the resistance to treatment	Recruiting	https://clinicaltrials.gov/study/NCT04866680 (accessed on 1 October 2023)
**ctDNA**	Observation prospective	Stage IIB, IIC melanoma or fully resectable Stage III B/C/D cutaneous melanoma.	28	To assess the feasibility of generating patient specific ctDNA assay from Signatera© test for primary melanoma samples submitted with clinical stage IIB/IIC and stage III melanoma patients.	Active not recruiting	https://clinicaltrials.gov/study/NCT05736523 (accessed on 1 October 2023)
**ctDNA**	Phase II/III	Stage IIB or IIC melanoma (sentinel lymph node (SNLB) staged) patients	8	Overall survival. To use ctDNA as a tool for indication of nivolumab adjuvant after resection of primary tumour.	Paused for protocol redesign	https://clinicaltrials.gov/study/NCT04901988 (accessed on 1 October 2023)
**ctDNA**	Prospective single group assignment	Patient with a metastatic choroidal melanoma	40	Assessment and development of circulating tumour DNA detection techniques	Completed	https://clinicaltrials.gov/study/NCT01334008 (accessed on 1 October 2023)
**ctDNA**	Prospective single group assignment	Uveal melanoma patients with hepatic metastasis eligible for surgery	60	Correlation between the circulating tumour DNA rate before/after surgery and the rate of effective complete resection	Completed	https://clinicaltrials.gov/study/NCT02849145 (accessed on 1 October 2023)
**ctDNA**	Prospective single group assignment	Uveal melanoma regardless of stage	800	To observe the prevalence of the ctDNA at the diagnostic and its evolution during 3 years.	Completed	https://clinicaltrials.gov/study/NCT02875652 (accessed on 1 October 2023)
**cfDNA**	Prospective single group assignment	Stage IV melanoma patients	22	To determine the mutational status in circulating DNA with the Sequenom mass array. Results obtained before and after treatment will be compared with the primary tumour genotype	Completed	https://clinicaltrials.gov/study/NCT02133222 (accessed on 1 October 2023)

**Table 2 cells-13-00019-t002:** List of emerging minimally invasive biomarkers in melanoma and their characteristics.

Biomarker	Characteristics
ctDNA quantification/mutation detection	Improved sensitivity/specificity compared to serological proteins; rapid integration into clinical practice but high cost; diminished validity in intracranial-only disease
ctDNA methylation signature	Exciting novel biomarker with tissue-independent capabilities; promising marker in intracranial disease
Circulating Tumour cells	Pivotal marker in the development of liquid biopsy hypothesis; standardised techniques are still a challenge
Extracellular Vesicles	Comprehensive biomarker that encapsulates both genomic and proteomic information; further standardisation is required regarding isolation and analytical methodology
Circulating MicroRNA and long non-coding RNA	Biomarker with an expanding discovery platform but no clear clinical indication yet; development of multi-RNA panels may be required
Intestinal microbiome	Non-invasive biomarker with rapidly gaining attention; consensus on reproducible, measurable readout is still indetermined

## Data Availability

Not applicable.

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
