# Peer review of "Promising and Minimally Invasive Biomarkers: Targeting Melanoma"

_cells, 2023, doi:10.3390/cells13010019_

Round 1

Reviewer 1 Report

Comments and Suggestions for Authors

The review is well-written and organized. A few points are following:

1) I would recommend to show the RNA of EVs not limited to microRNA, therefore indicating both coding and non-coding RNAs.
The legend of Figure 1 must include EVs in the description.

2) Add a reference in raw 530 that justifies 10 micrometers as size of putative extracellular vesicles.

3) Raw 534: pleaase substitute the word "developmental" with "biogenetic".

4) Exosomes are generated through the recycling of the multivesicular bodies, therefore not necessarily as a consequence of endocytic processes only. Please adjust it in the text.

5) The results of terminated clinical trials should be all indicated in the main text.

Author Response

Reviewer 1:

The review is well-written and organized. A few points are following:

1) I would recommend to show the RNA of EVs not limited to microRNA, therefore indicating both coding and non-coding RNAs.

  • To address this comment, we added further text between lines 552-557, to clarify that EV mediated RNA cargo can be both coding and non-coding RNA. We reinforced this on the title of the respective paragraph.

The legend of Figure 1 must include EVs in the description

  • This has not been corrected as extracellular vesicles are mentioned in full word (and not as an acronym) on the figure, therefore an explanation in the legend is not required at this point.

2) Add a reference in raw 530 that justifies 10 micrometers as size of putative extracellular vesicles

  • References 169 (Robbins et al, 2014) and 170 (Raposo et al, 2013) have been added to support the statement on line 539 (previous line 530). We thank the reviewer for this suggestion.

3) Raw 534: please substitute the word "developmental" with "biogenetic".

  • This has now been substituted in line 543 (this was previously line 534).

4) Exosomes are generated through the recycling of the multivesicular bodies, therefore not necessarily as a consequence of endocytic processes only. Please adjust it in the text.

  • We thank the reviewer for this insight. We have added additional text in lines 543-545 to this effect.        

5) The results of terminated clinical trials should be all indicated in the main text

  • There were three terminated studies in Table 1. We have provided explanation for the study by Van Der Hiel et al in lines 281-285. We decided to remove study NCT02439008 (Early Biomarkers of Tumor Response in High Dose Hypofractionated Radiotherapy Word Package 3: Immune Response) given the lack of information about its termination. Lastly, the 3rd study (NCT04901988) was not fully terminated but rather paused for protocol redesign, therefore we decided to keep in on Table 1 and changed its status to paused (this is in page 6).

Reviewer 2 Report

Comments and Suggestions for Authors

This review covers a wide range of melanoma biomarkers for liquid biopsy, from the genomic classification of melanoma to the latest developments in minimally invasive biomarkers like ctDNA, CTCs, EVs, gut microbiota. This comprehensive approach ensures that readers get a holistic understanding of the current state of melanoma research and treatment. I support the publication of this review after minor revision.

While the authors gave a comprehensive discussion of the application of ctDNA for monitoring melanoma, they lacked a discussion of other circulating free nucleotides, such as miRNA and long noncoding RNA. The authors can add a section on them.

Author Response

Reviewer 2:

This review covers a wide range of melanoma biomarkers for liquid biopsy, from the genomic classification of melanoma to the latest developments in minimally invasive biomarkers like ctDNA, CTCs, EVs, gut microbiota. This comprehensive approach ensures that readers get a holistic understanding of the current state of melanoma research and treatment. I support the publication of this review after minor revision.

While the authors gave a comprehensive discussion of the application of ctDNA for monitoring melanoma, they lacked a discussion of other circulating free nucleotides, such as miRNA and long noncoding RNA. The authors can add a section on them.

  • We thank Reviewer 2 for this valuable suggestion. To address this in our manuscript, we have added paragraph between lines 667-689 with a special mention to miRNA, which we are hoping it provides a succinct but adequate view on the most pertinent research on this matter. Please note that citations 201-205 have also been added. We also wish to add that we acknowledge the Reviewer’s comment that the portion of ctDNA occupies more space in our review, as compared to other circulating free nucleotides, such as miRNA and long coding RNA. This is explained by the recent explosion of research on the field of ctDNA, especially in melanoma, and its rapid integration into clinical practice, which we thought worthy of extensive coverage in the manuscript, as compared with the less researched space of miRNA and long noncoding RNA.

Reviewer 3 Report

Comments and Suggestions for Authors

1. Figure 1 shows intestinal microbiome" and "taxonomic diversity" but does not explain what these are or why they are relevant to the rest of the figure, and neither does the text. 

2. The distracted text in the Table makes it hard to read 

3. While there may not be plagiarism in the paper much of the text is paragraph after paragraph of minutiae that is probably right out of the references and makes the text hard to read. A table summarizing potential future biomarkers would really help, especially if it could be correlated with Figure 2. 

Comments on the Quality of English Language

Some tortuous grammar 

Author Response

Reviewer 3:

1.Figure 1 shows “intestinal microbiome" and "taxonomic diversity" but does not explain what these are or why they are relevant to the rest of the figure, and neither does the text.

  • Intestinal microbiome is now explained on line 736 and taxonomic diversity on lines 739-741. Intestinal microbiome/taxonomic diversity were added to Fig 1 as emerging minimally invasive biomarkers of prognostic and predictive value in melanoma. This is further explained in the last section of the manuscript.
  1. The distracted text in the Table makes it hard to read 
  • We thank the reviewer for their comment; we have adjusted the table text format accordingly.
  1. While there may not be plagiarism in the paper much of the text is paragraph after paragraph of minutiae that is probably right out of the references and makes the text hard to read. A table summarizing potential future biomarkers would really help, especially if it could be correlated with Figure 2. 
  • We took the Reviewer’s comment into consideration and added a summarizing table (Table 2; page 21).

Round 2

Reviewer 3 Report

Comments and Suggestions for Authors

none